# Role of Tumor-Associated Macrophages in Sarcomas

**DOI:** 10.3390/cancers13051086

**Published:** 2021-03-03

**Authors:** Tomohiro Fujiwara, John Healey, Koichi Ogura, Aki Yoshida, Hiroya Kondo, Toshiaki Hata, Miho Kure, Hiroshi Tazawa, Eiji Nakata, Toshiyuki Kunisada, Toshiyoshi Fujiwara, Toshifumi Ozaki

**Affiliations:** 1Department of Orthopaedic Surgery, Okayama University Graduate School of Medicine, Dentistry and Pharmaceutical Sciences, Okayama 700-8558, Japan; akysda@gmail.com (A.Y.); me20034@s.okayama-u.ac.jp (H.K.); toshiakih1080@gmail.com (T.H.); miho.suzuki0000@gmail.com (M.K.); eijinakata8522@yahoo.co.jp (E.N.); toshi-kunisada@umin.ac.jp (T.K.); tozaki@md.okayama-u.ac.jp (T.O.); 2Department of Surgery, Orthopaedic Service, Memorial Sloan Kettering Cancer Center, New York, NY 10065, USA; healeyj@MSKCC.ORG (J.H.); ogura-tky@umin.ac.jp (K.O.); 3Department of Gastroenterological Surgery, Okayama University Graduate School of Medicine, Dentistry and Pharmaceutical Sciences, Okayama 700-8558, Japan; htazawa@md.okayama-u.ac.jp (H.T.); toshi_f@md.okayama-u.ac.jp (T.F.); 4Center for Innovative Clinical Medicine, Okayama University Graduate School of Medicine, Dentistry and Pharmaceutical Sciences, Okayama 700-8558, Japan

**Keywords:** sarcoma, tumor-associated macrophage, prognosis, clinical trial, immunotherapy

## Abstract

**Simple Summary:**

Recent studies have shown the pro-tumoral role of tumor-associated macrophages (TAMs) not only in major types of carcinomas but also in sarcomas. Several types of TAM-targeted drugs have been investigated under clinical trials, which may represent a novel therapeutic approach for bone and soft-tissue sarcomas.

**Abstract:**

Sarcomas are complex tissues in which sarcoma cells maintain intricate interactions with their tumor microenvironment. Tumor-associated macrophages (TAMs) are a major component of tumor-infiltrating immune cells in the tumor microenvironment and have a dominant role as orchestrators of tumor-related inflammation. TAMs promote tumor growth and metastasis, stimulate angiogenesis, mediate immune suppression, and limit the antitumor activity of conventional chemotherapy and radiotherapy. Evidence suggests that the increased infiltration of TAMs and elevated expression of macrophage-related genes are associated with poor prognoses in most solid tumors, whereas evidence of this in sarcomas is limited. Based on these findings, TAM-targeted therapeutic strategies, such as inhibition of CSF-1/CSF-1R, CCL2/CCR2, and CD47/SIRPα, have been developed and are currently being evaluated in clinical trials. While most of the therapeutic challenges that target sarcoma cells have been unsuccessful and the prognosis of sarcomas has plateaued since the 1990s, several clinical trials of these strategies have yielded promising results and warrant further investigation to determine their translational benefit in sarcoma patients. This review summarizes the roles of TAMs in sarcomas and provides a rationale and update of TAM-targeted therapy as a novel treatment approach for sarcomas.

## 1. Introduction

Sarcomas, which are broadly categorized as bone sarcomas or soft-tissue sarcomas, represent a clinically and molecularly heterogeneous group of mesenchymal malignancies with more than 50 histological subtypes [1,2,3]. While wide surgical resection is a mainstay of treatment for sarcomas, multidisciplinary treatments with multiagent chemotherapy and/or radiotherapy are performed according to the histological diagnosis. The introduction of systemic chemotherapy in the 1970s–1980s substantially improved the prognosis of patients with osteosarcoma, which is the most common primary bone sarcoma [4,5,6]. Since then, enormous effort has been made to develop novel drugs; however, effective therapies have not emerged, and the prognosis for sarcomas has plateaued since the 1990s [4]. For soft-tissue sarcomas such as myxoid liposarcoma and synovial sarcoma, doxorubicin is the only drug that has been demonstrated to be effective for survival, but the benefit is limited. Some molecular targeted drugs, such as pazopanib, trabectedin, and eribulin, have been recently approved by the U.S. Food and Drug Administration (FDA), but these therapies do not have a substantial cure rate [7].

Most of the therapeutic challenges targeting sarcoma cells have failed. Unfortunately, recent introduction of emerging anti-PD-1 immunotherapy to treat sarcomas has resulted in a poor response [8,9]. An alternative strategy is to target cells in the sarcoma microenvironment. Tumor-associated macrophages (TAMs) are major components in the tumor microenvironment and have a dominant role as orchestrators of cancer-related inflammation [10,11,12]. Several preclinical approaches targeting TAMs or inhibiting their tumor-promoting functions have been successful and are regarded as promising therapeutic strategies following the development of immune checkpoint inhibitors [13,14,15,16,17,18,19,20]. In this review, we summarize the roles of TAMs and their clinical relevance in sarcomas, and provide an update on recent therapeutic advancements targeting sarcoma TAMs. We searched articles published until December 2020 in PubMed using the following terms: “macrophage,” “tumor-associated macrophage,” “sarcoma,” “bone sarcoma,” “soft-tissue sarcoma,” and histological diagnosis terms, such as “osteosarcoma,” “Ewing sarcoma,” “chondrosarcoma,” “leiomyosarcoma,” “liposarcoma,” “undifferentiated sarcoma,” “synovial sarcoma,” and “dermatofibrosarcoma protuberans,” in various combinations. Abstracts of the manuscripts in English were reviewed for relevance. Studies reporting the prognostic value of TAMs in bone and soft-tissue sarcomas were all included. Finally, we searched ClinicalTrials.gov for clinical trials with TAM-targeted drugs.

## 2. Role of Tumor-Associated Macrophages (TAMs)

Among the innate and adaptive immune cells that are recruited to the tumor microenvironment, macrophages are particularly abundant and influence tumor growth and progression [10,11,12]. There are multiple precursors of TAMs, including circulating blood monocytes, monocyte-related myeloid-derived suppressor cells (M-MDSCs), and tissue-resident macrophages [11]. They are recruited to the tumor sites in response to cytokines (such as colony stimulating factor (CSF)-1, interleukin (IL)-34, and members of the vascular endothelial growth factor (VEGF) family) and chemokines (such as chemokine C-C motif ligand 2 (CCL2) and CCL5) and differentiate into TAMs [11]. In general, monocytes/macrophages can be polarized to M1-like (classically activated) or M2-like (alternatively activated) macrophages [12]. TAMs typically display a pro-tumorigenic phenotype associated with the M2-like profile, whereas the anti-tumorigenic function is associated with the M1-like phenotype [21].

TAMs play specialized functional roles in tumor progression, including cancer progression, metastasis, angiogenesis, and immune suppression (Figure 1) [12]. TAM-derived IL-6 and mitogens promote the occurrence and development of hepatocellular carcinoma via activation of signal transducer and activator of transcription 3 (STAT3) and nuclear factor kappa B (NF-κB) signaling, respectively, which also promote resistance to chemotherapy [22,23]. Similarly, the presence of TAM-derived inflammatory cytokines IL-23 and IL-17 are associated with tumor progression [24]. TAM expressions of IL-6 and tumor necrosis factor (TNF)-α promote resistance to chemotherapy and targeted agents [25]. Several TAM-produced proteases, including cathepsin B, matrix metalloproteinase (MMP)-2, MMP-7, MMP-9, and the extracellular matrix (ECM), contribute to tumor invasion and metastasis [26]. Tumor angiogenesis is promoted by the VEGF, TNF-α, IL-1β, IL-8, platelet-derived growth factor (PDGF), basic fibroblast growth factor (bFGF), and MMPs, which are produced by TAMs [27]. TAMs can also promote the immunosuppressive activity of regulatory T cells through IL-10 and transforming growth factor (TGF)-β [10,28]. TAMs often express programmed death-ligand 1 (PD-L1)/L2, B7-H4, and V-domain Ig suppressor of T cell activation (VISTA), which trigger the inhibitory PD-1-mediated immune checkpoint in T cells [11,29,30].

Regarding the clinical significance of TAMs, a high density of M2-like TAMs in the tumor microenvironment is associated with a poor survival outcome in many types of malignant tumors [11,31,32]. The common markers for M1-like TAMs in human samples are human leucocyte antigen (HLA)-DR, inducible nitric oxide synthase (iNOS), and pSTAT1. On the other hand, common markers for M2-like TAMs are CD163, CD204, and CD206, attributable to the high expression of the mannose receptor-1 (CD206) and macrophage scavenger receptors (CD163 and CD204) by the M2-like TAMs [12]. In patients with breast cancer, a high density of CD163^+^ macrophages has been associated with poor histological grade, hormonal receptor negativity, lymph node metastasis, and poor survival outcome [33]. Zhang et al. developed a meta-analysis with 55 studies, where they evaluated the correlation between the TAM infiltration and clinical prognosis. They showed the adverse effects of TAMs on survival in breast, gastric, bladder, ovarian, oral, and thyroid cancer patients in their results. In contrast, they observed positive effects of TAMs on survival in patients with colorectal cancer [32]. Evidence suggests that increased inflammation-related gene expressions, especially those related to the polarization of macrophages, are also associated with poor survival [16]. However, there are conflicting data for several types of cancers, such as stomach and prostate cancer [32]. These are possibly related to the type of analysis performed (e.g., quantitation of stromal versus intratumoral macrophages), the stage of cancer evaluated, or the use of different macrophage markers.

## 3. Clinical Relevance of the Infiltration of TAMs in Bone Sarcomas

### 3.1. Osteosarcoma

Osteosarcoma is the most common primary malignancy of bone and commonly arises in adolescent and young adult populations [34,35]. It is characterized histologically by the production of osteoid by malignant cells and has a variety of histological subtypes, including conventional (osteoblastic, chondroblastic, and fibroblastic types), telangiectatic, small cell, low-grade central, parosteal, periosteal, high-grade surface, and secondary osteosarcoma [3].

The clinical relevance of TAMs in osteosarcoma was first demonstrated by Buddingh et al. [36] (Table 1). Gene profiling analysis of biopsies performed on non-metastatic versus metastatic osteosarcomas revealed a high expression of macrophage-associated genes, such as CD14 and HLA-DRα, in non-metastatic tumors, which was expressed by infiltrating hematopoietic cells [36]. The total number of macrophages, which was determined based on the number of CD14^+^ macrophages, was associated with better survival, but that of the M1-phenotype (CD14/HLA-DRα) and the M2-phenotype (CD14/CD163) were not correlated with prognosis [36]. A high number of CD14^+^ macrophages in prechemotherapy samples were associated with a better response to neoadjuvant chemotherapy, and their numbers increased after chemotherapy [36].

Controversy exists regarding the prognostic significance of the M1/M2-phenotype in osteosarcoma. Dumars et al. compared the expressions of several molecules, including TAM markers, CD146 (vascularity), and osteoprotegerin in localized and metastatic tumors by using immunohistochemistry [37]. A higher infiltration of iNOS^+^ M1-like TAMs was observed in localized tumors, whereas a higher vascular density (CD146^+^ cells), which was associated with CD163^+^ M2-like TAMs, was observed in metastatic tumors [37]. Dumars et al. concluded that a dysregulation of M1/M2 polarization in favor of M1-like TAMs is associated with localized osteosarcoma [37]. On the contrary, the correlation between the M2-phenotype and worse prognosis was demonstrated by Gomez-Brouchet et al. [38]. In an immunohistochemical analysis of 124 pretherapeutic biopsies, 43.8% and 23.4% had CD163^+^ and CD68^+^ staining greater than 50% per core, respectively [38]. A high level of CD163 staining was associated with a high expression level of the M2-marker CMAF (macrophage activation factor) but was not related to a high expression level of the M1-marker pSTAT1 [38]. In terms of prognostic relevance, a high level of CD163^+^ cells in the biopsies was significantly correlated with a higher overall and metastasis-free survival, whereas a trend for a higher survival was observed in patients with >50% CD68^+^ cells [38].

The clinical significance of CD47 and signal-regulatory protein α (SIRPα), which are macrophage-related checkpoints, was recently reported. CD47, a transmembrane protein found ubiquitously expressed on normal cells, has increased its expression in a high proportion of malignant tumor cells. This protein acts primarily as a dominant “do not eat me” signal [42,43]. If the tumor cells express CD47, it binds to SIRPα on phagocytic immune cells, preventing engulfment [42,43,44]. Dancsok et al. investigated the expressions of CD68, CD163, CD47, and SIRPα for 1242 sarcomas (24 histological subtypes) by immunohistochemical analysis [39]. Among 247 patients with osteosarcoma, the median numbers of CD68^+^ and CD163^+^ macrophage infiltrations were approximately 110/mm^2^ and 150/mm^2^, respectively, which were lower than those of angiosarcoma, undifferentiated pleomorphic sarcoma, dedifferentiated liposarcoma, myxofibrosarcoma, pleomorphic liposarcoma, and leiomyosarcoma. CD47 expression was observed in approximately 53% of osteosarcoma cells, whereas SIRPα expression was identified in approximately 32% of the infiltrating macrophages [39]. Interestingly, the CD47 and SIRPα expression levels were correlated with higher CD68^+^ and CD163^+^ macrophage infiltrates [39]. Although the prognostic relevance of CD68^+^ and CD163^+^ macrophage infiltration in osteosarcoma was not described, lower SIRPα levels were associated with a worse overall survival among non-translocation sarcomas, and CD47 expression turned out to be a poor prognostic factor in osteosarcomas [39].

### 3.2. Ewing’s Sarcoma

Ewing’s sarcoma is the second most frequent bone tumor of childhood and adolescence and is characterized by specific chromosomal translocations that produce *FET-ETS* fusion oncogenes such as *EWS-FLI1* [45]. Despite its unknown mechanisms, increased white blood cell (WBC) counts, elevated C-reactive protein (CRP) concentration, and increased erythrocyte sedimentation rate (ESR) are frequently observed. These biological characteristics suggest the involvement of TAMs in the initiation and maintenance of Ewing’s sarcomas.

A close correlation between tumor-related inflammation and the infiltration of TAMs in the tumor microenvironment was reported by Fujiwara et al. [40] (Table 1). A total of 41 patients with Ewing’s sarcoma were divided into two groups according to the density of CD68^+^ macrophages: 21 patients (51%) with lower infiltration (≤30 CD68^+^ cells/HPF) and 20 patients (49%) with higher infiltration (>30 CD68^+^ cells/HPF) [43]. A higher extent of TAM infiltration, greater microvascular density, elevated white blood cell (WBC) counts (>6800 cells/μL) and C-reactive protein (CRP) values (>0.2 mg/dL) were significantly associated with worse prognosis [40]. In addition, higher TAM infiltration was also associated with elevated WBC counts and CRP values, as well as higher microvascular density, and this turned out to be an independently poor prognostic factor [40]. On the other hand, Handl et al. showed no correlation between CD68^+^ macrophage density and clinicopathological parameters [41]. However, the higher number and density of CD163^+^ TAMs were correlated with localized disease but there was a trend toward more prolonged survival in relation to a higher density of CD163^+^ TAMs [41]. In this study, 71% and 79% of 24 cases showed a modest to massive infiltration of CD68^+^ and CD163^+^ cells, respectively [41]. Further analyses based on the larger cohorts are necessary to clarify the prognostic role of CD68^+^ and CD163^+^ macrophage density in Ewing’s sarcoma.

## 4. Clinical Relevance of the Infiltration of TAMs in Soft-Tissue Sarcomas

### 4.1. Leiomyosarcoma

Leiomyosarcoma is a malignant soft-tissue tumor derived from the smooth muscle lineage [46]. It most often develops in the retroperitoneum but can also develop in the extremities [46]. Previous publications have indicated the correlation between the M2-like TAMs or M2-related markers and worse prognosis in leiomyosarcoma (Table 2) [47,48]. Global gene expression profiling by Lee et al. revealed high expressions of several macrophage-associated genes, including CD68 and CD163, which was confirmed by immunohistochemistry [47]. High densities of CD68^+^ or CD163^+^ macrophages were associated with a worse disease-specific survival in non-gynecologic leiomyosarcoma, whereas there was no association in gynecologic leiomyosarcoma [47]. The 5-year disease-specific survival was <40%, 79%, and 100% in patients with non-gynecologic leiomyosarcomas showing dense, moderate, and sparse CD163^+^ macrophages, respectively [47]. The clinical significance of CSF-1-associated proteins (CD163, CD14, and cathepsin L) was investigated by Ganjoo et al. [48]. The increased levels of CD16 and cathepsin L were both associated with a worse prognosis in gynecologic leiomyosarcoma [48]. In addition, positive staining of all three markers resulted in poor overall survival, which was not confirmed in non-gynecologic leiomyosarcoma [48]. These data indicated the pro-tumoral role of TAMs and CSF-1-associated proteins in leiomyosarcoma [48].

### 4.2. Myxoid Liposarcoma

Myxoid liposarcoma, which is the second most common subtype of liposarcoma, occurs predominantly in the extremities of young adults and has a high tendency to metastasize to soft tissue [54]. This tumor is characterized by translocations producing *FUS-DDIT3* or, rarely, *EWSR1-DDIT3* fusion transcripts [55,56]. Correlation between the infiltration of M2-like TAMs and poor prognosis was described by Nabeshima et al. [49] (Table 2). Clinicopathologic evaluation using immunohistochemistry for CD68 and CD163 revealed that a greater infiltration of either CD68^+^ macrophages or CD163^+^ M2-like TAMs was associated with decreased overall survival [49]. Interestingly, the macrophage-conditioned medium stimulated cellular motility and invasion by activating the epidermal growth factor receptor (EGFR) with the ligand that was suggested to be an epidermal growth factor (EGF)-like growth factor (HB-EGF) [49]. Thus, Nabeshima et al. concluded that TAMs, HB-EGF, and EGFR could be new candidates for therapeutic targets of myxoid liposarcoma [49].

### 4.3. Synovial Sarcoma

Synovial sarcoma (SS) is a common soft-tissue sarcoma that occurs in adolescents and young adults [57]. This sarcoma is characterized by a specific chromosome translocation that produces the *SS18-SSX1/2/4* fusion gene [58]. A correlation between M2-like TAM infiltration and immune cell infiltration in the tumor microenvironment was reported by Oike et al. (Table 2) [50]. In a screening of macrophage-related, lymphocyte-related, and immune checkpoint markers, all tumors (*n* = 36) had CD163^+^ macrophages, whereas 72%, 92%, and 75% were positive for CD4, CD8, and forkhead box P3 (FOXP3), respectively [50]. Interestingly, there was a correlation between the number of CD163^+^ macrophages and the densities of CD8^+^ and FOXP3^+^ lymphocytes [50]. Patients with lower levels of CD163^+^ macrophage infiltration, high levels of CD8^+^, and FOXP3^+^ lymphocyte infiltration were associated with better overall survival [50]. Regulatory T cells promote evasion of cancer cells from immune responses and often contribute to worse survival [59,60]. Thus, the correlation between a high infiltration of FOXP3^+^ lymphocytes and a better survival outcome in SS was different from the former publications [50]. This inconsistency may be due to their choice of cut-off value and their assumption that FOXP3^+^ lymphocytes that infiltrate into the tumor microenvironment in SS might represent a subset other than regulatory T cells [50]. FOXP3^+^ T cells comprise functionally different subsets, including non-regulatory T cells. Further studies are needed to determine the role of FOXP3^+^ lymphocytes in SS.

### 4.4. Dermatofibrosarcoma Protuberans

Dermatofibrosarcoma protuberans is a cutaneous fibroblastic tumor that is locally aggressive, with a tendency for local recurrence but rarely metastasizes [61]. Although the prognostic significance of M1/M2-like TAMs were not described, a correlation between M2-like TAMs and the local aggressiveness of this tumor was reported (Table 2) [51]. Immunohistochemical staining revealed CD163, CD206, and periostin, which recruits M2-TAMs in glioblastoma multiforme, and the expressions of these markers have been identified in all studied cases (*n* = 10) at the peripheral areas of the tumors [51]. MMP1 and MMP12, which are modulated by periostin, were also observed in the TAMs-detected area, which may indicate the local aggressiveness of dermatofibrosarcoma protuberans [51].

### 4.5. Undifferentiated Pleomorphic Sarcoma

Undifferentiated pleomorphic sarcoma (UPS), formerly known as malignant fibrous histiocytoma, is one of the most common soft-tissue sarcomas, classified as a subcategory of undifferentiated sarcoma [3]. This tumor is characterized by a high rate of local recurrence (13–42%) and distant metastasis (31–35%) due to its infiltrative nature [62]. The infiltration of M2-like TAM is strongly associated with worse outcomes in UPS patients (Table 2) [52]. In a screening of macrophage-related markers by immunohistochemical staining, Komohara et al. reported that the densities of Iba1^+^ (M1/M2), CD163^+^ (M2), and CD204^+^ (M2) TAMs were positively correlated with each other, and that the density of CD163^+^ TAMs was high in older patients or those with a smaller tumor size [52]. In a clinicopathologic analysis of 24 patients, there was a trend toward worse survival outcomes in patients with a high density of CD163^+^ or CD204^+^ TAMs [52]. Shiraishi et al. from the same group validated the correlation between a high percentage of CD163^+^ TAMs and worse survival outcomes and found a positive correlation with a high tumor grade [53]. Of note, TAM-induced cell proliferation was observed in leiomyosarcoma and myxofibrosarcoma cell lines, and was promoted by IL-6 secreted from TAMs [53].

UPS is known to have increased infiltration of CD68^+^ macrophages and CD163^+^ TAMs among the various subtypes of sarcoma [39]. Dancsok et al. reported that pleomorphic types demonstrated the highest counts of both CD68^+^ and CD163^+^ macrophages, particularly in UPS (median CD68 = 460/mm^2^, CD163 = 512/mm^2^) [39]. Regarding the macrophage polarization, a higher proportion of M2-like macrophages than M1-like macrophages was observed, particularly in UPS (adjusted mean CD163/CD68 = 20.7) and leiomyosarcoma (adjusted mean CD163/CD68 = 17.4) [39]. The prognostic significance of the macrophage polarization in UPS was not described in this paper [39].

## 5. Therapeutic Trials Targeting TAMs in Sarcomas

The identification of the various functions of TAMs has provided unprecedented opportunities for the development of novel therapies for malignant diseases. Accumulating evidence of the pro-tumoral roles of TAMs indicates the antitumor effect of TAM-targeted therapies. Therapeutic strategies directed at TAMs can be classified into four types: (1) limiting macrophage recruitment (e.g., CCL2/C-C motif receptor 2 (CCR2) inhibition), (2) reprogramming TAMs into antitumor macrophages (e.g., CSF-1/CSF-1R inhibition), (3) targeting the activation of TAMs (e.g., nucleotide-binding oligomerization domain-containing protein 2 (NOD2)/NF-κB), and (4) activation of macrophage phagocytosis (e.g., CD47/SIRPα inhibition) (Figure 2).

Although several drug interventions have been employed in clinical trials for major types of cancer, trials for bone and soft-tumor sarcomas are limited to date.

Pexidartinib (PLX3397), which is a potent CSF-1/CSF-1R inhibitor, has been tested on sarcomas as part of a clinical trial (NCT01004861, NCT01525602, NCT02390752, and NCT02584647). CSF-1, which is highly expressed in several types of solid tumor, plays a significant role in the recruitment of peripheral blood monocytes to the tumor microenvironment, differentiation into macrophages, and polarization of macrophages toward an M2-like phenotype via binding to CSF-1R, which is a tyrosine kinase receptor that is highly expressed in circulating monocytes and macrophages [11,12,25,63,64]. Preclinical studies have shown that the CSF-1/CSF-1R signaling cascade not only decreases the number of infiltrating TAMs but also repolarizes M2-like to M1-like phenotypes within the tumor microenvironment [13,65]. PLX3397 was designed to stabilize CSF-1R in the auto-inhibited state by interacting with the CSF-1R juxtamembrane region, thus resulting in the inactivation of the kinase domain and the prevention of CSF-1 and adenosine triphosphate (ATP) binding [66]. The oral administration of PLX3397 (Turalio^®^) has been approved by the U.S. Food and Drug Administration (FDA) for the treatment of unresectable tenosynovial giant cell tumor, which is a rare and locally aggressive non-malignant tumor that overexpresses CSF-1 [67,68,69]. This drug is currently under investigation with sirolimus in a phase I/II trial for unresectable sarcomas, including Ewing’s sarcoma, liposarcoma, leiomyosarcoma, malignant peripheral nerve sheath tumor, synovial sarcoma, and rhabdomyosarcoma (NCT02584647).

Mifamurtide, which is also known as liposomal muramyl tripeptide and muramyl tripeptide phosphatidylethanolamine (Mepact^®^), is indicated in children, adolescents, and young adults for the treatment of high-grade, resectable non-metastatic osteosarcoma in the EU [70]. This drug is intravenously administered in conjunction with postoperative multiagent chemotherapy. Mifamurtide stimulates immune responses by binding to NOD2 in an intracellular pattern-recognition receptor molecule expressed mainly in monocytes, macrophages, and dendritic cells [71,72,73]. Binding to NOD2 results in the activation of the NF-kB pathway, which leads to an increased production of pro-inflammatory cytokines, such as TNF-a, IL-1, IL-6, IL-8, interferon gamma (IFN-gamma), and the serum CRP [71,72,74]. Mifamurtide also activates NLR family pyrin domain-containing 3 (NLRP3), which is an essential component of inflammasome, which is a protein complex that promotes the cleavage of procaspase 1 in its active form [72]. Active caspase 1 is essential for the activation of pro-inflammatory cytokines like IL-1β. Overall, these pathways in macrophages and monocytes contribute to inflammation, the release of antimicrobial peptides, dendritic cell recruitment, the polarization of T-helper cells, and promotion of bactericidal and potential tumoricidal effects [72]. A phase III randomized clinical trial was conducted by the Children’s Oncology Group from 1993 to 1997 [75]. Significant improvements in event-free survival and overall survival were observed in patients who received mifamurtide [73,75,76]. However, in the U.S., this drug is an investigational agent, because the FDA refused to approve mifamurtide due to insufficient evidence of a survival advantage [71].

Trabectedin (Yondelis^®^), which was recently approved in the treatment of advanced soft-tissue sarcomas, was reported to partially deplete circulating monocytes and TAMs [11,77,78]. This was observed in patients with cancer that showed delayed, persistent responses to trabectedin. Trabectedin has been shown to activate a TNF-related apoptosis-inducing ligand (TRAIL)/caspase 8-dependent pathway of apoptosis [11,79]. Of note, monocytes are sensitive to TRAIL, because they express low levels of TRAIL decoy receptors [80]. Germano et al. reported that trabectedin reduced TAM density, which was associated with decreased angiogenesis in mouse tumor models and in human sarcoma specimens [78]. These findings raised the issue for an exploitation of the combined use of trabectedin with anti-angiogenic agents and/or immunotherapeutic drugs.

The other candidate targets of TAMs include CCL2/CCR2, CD40, and CD47/SIRPα [11,12,81]. Regarding the CCL2/CCR2 axis, which plays a role in the recruitment in tumors, phase I and II clinical trials of anti-CCL2 antibodies (carlumab) were performed and completed in solid tumors (NCT00992186) [82] and metastatic prostate cancer (NCT01204996) [83], respectively, which showed no significant antitumor activity as a single agent. The phase I trial of the CCR2 antagonist (PF-04136309) in advanced pancreatic adenocarcinoma (NCT01413022) confirmed its safety and tolerability in an objective tumor response [84]. CD40 is a surface marker of macrophages, which is highly expressed on M1-like TAMs [85]. The combination of a CD40 agonist with gemcitabine effectively circumvented tumor-mediated immune suppression by promoting antitumor macrophages, which increased the survival rate in patients with surgically incurable pancreatic ductal adenocarcinoma (PDAC) [86]. Several phase I trials of the anti-CD40 agonist have shown tolerability (NCT01433172) [87,88]. CD47, an integrin-associated protein that is overexpressed in malignant tumor cells, functions as an inhibitor of macrophage-mediated phagocytosis through the ligation of SIRPα [89]. Several drugs targeting CD47/SIRPα were investigated in several clinical trials. A phase I trial of intratumoral TTI-621, a SIRPα-Fc fusion protein, showed tolerability and moderate antitumor efficacy in patients with T cell lymphoma [90]. The intravenous administration of ALX148, which binds to CD47, is under investigation in combination with trastuzumab or pembrolizumab (NCT03013218). The safety and efficacy of these drugs remain to be investigated in patients with sarcoma. The angiopoietin receptor TIE2, a molecule previously known to be restricted to endothelial and hematopoietic stem cells [91,92], is expressed on a subset of TAMs [93]. TIE2^+^ TAMs are closely associated with tumor vasculature and have been found crucial for angiogenesis, which depends on angiopoietin-2 (Ang2), a TIE2 ligand produced by endothelial cells. Several drugs designed to target the Ang2–TIE2 axis, such as trebananib and venucizumab, have been tested for solid tumors [94,95]. Trebananib was combined with paclitaxel, trastuzumab, or bevacizumab, which was tolerable but, so far, its efficacy has been limited in patients with breast cancer [96]. Venucizumab has an acceptable safety and tolerability profile for a heterogeneous group of advanced solid tumors [97], but combined use with mFOLFOX-6 showed limited clinical benefit for metastatic colorectal cancer [98]. The PD-1/PD-L1 axis has been an attractive target in cancer immunotherapy. PD-1 expression on TAMs correlates negatively with phagocytic potency against tumor cells [99], raising a possible effect of the combination of macrophage-targeted therapy and immune checkpoint modulation. A preclinical study demonstrated that the combination of a CSF-1R inhibitor with PD-1 or CTLA4 antagonists elicited tumor regression, while the single use of PD-1 or CTLA4 inhibitors showed limited efficacy [16]. However, a phase I/IIa trial of PLX3397 with pembrolizumab, an anti-PD-1 antibody, for advanced melanoma and other solid tumors was recently terminated for insufficient evidence of clinical efficacy (NCT02452424).

## 6. Conclusions and Future Perspectives

TAMs have important pro- and antitumor functions within the microenvironment of major types of cancers. However, the investigation of the clinical significance of TAMs in bone and soft-tissue sarcomas has been limited. Studies suggest that the infiltration of TAMs is associated with a worse survival outcome in patients with soft-tissue sarcomas, whereas there is no consensus in those with bone sarcomas. This inconsistency may be related to the type of analysis performed, the stage of the tumor, or the use of different macrophage markers. Further study based on a larger cohort with a protocol that reaches consensus is necessary to determine the clinical significance of TAMs in the microenvironment of bone sarcomas.

Given the association between the increased infiltration of TAMs and worse survival outcomes in most sarcoma subtypes, these cells represent promising targets for novel therapies. TAM-targeted therapeutic approaches have entered the clinical arena. Several types of TAM-targeted therapies, including CSF-1/CSF1-R inhibition, CCL2/CCR2 inhibition, and CD47/SIRPα targeting, have been investigated under clinical trials for major types of cancer. Among these, PLX3397 has been investigated in a single and combined use with sirolimus under clinical trials for a variety of histological subtypes of sarcoma. In preclinical studies, the systemic administration of PLX3397 resulted in the depletion of TAMs and changed the immune cell composition in the microenvironment of major cancer types [16,100,101]. In addition, PLX3397 was shown to decrease resistance to chemotherapy and radiotherapy [25,102,103], which indicates a promising antitumor effect for sarcomas that require multidisciplinary treatment.

The presumed hurdle of targeting TAMs could be drug toxicity, since the systemic depletion of macrophages may lead to increased infections or impaired tissue-resident macrophages [104]. In a phase II trial of PLX3397 for tenosynovial giant cell tumor, treatment-emergent adverse events resulted in permanent treatment discontinuation in 13% of PLX3397 recipients, in whom most of these adverse events were hepatotoxicity [68,105]. These findings underscore the necessity for the development of a novel approach that reduces the toxicity of TAM-targeted drugs. The recent development of nanomedicine strategies targeting TAMs are encouraging [106,107,108] and are key to overcoming the toxicity associated with the TAM-targeted therapeutics.

## Figures and Tables

**Figure 1 cancers-13-01086-f001:**
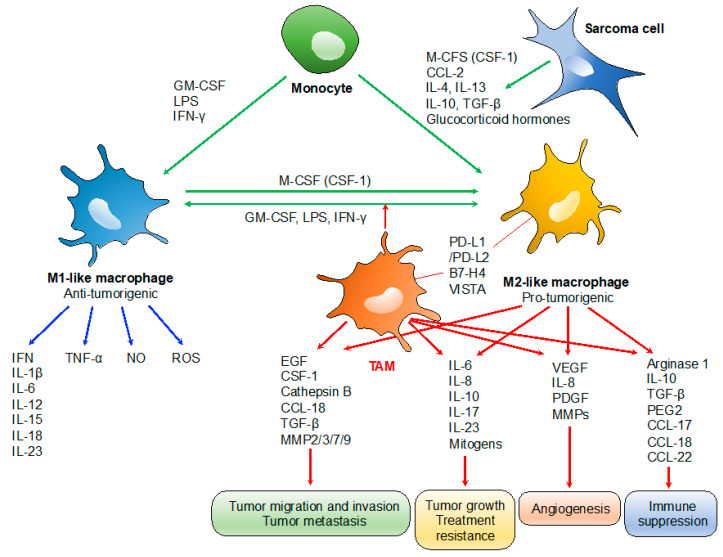
Roles of tumor-associated macrophages (TAMs) in tumor progression. TAMs typically display a pro-tumorigenic phenotype associated with the M2-like profile. Tumor growth: TAM-derived IL-6/IL-17/IL-23 and mitogens promote tumor growth and treatment resistance via activation of STAT3 and NF-κB) signaling, respectively. Tumor migration, invasion and metastasis: TAM-derived proteases such as cathepsin B, MMP-2, -3, -7, and -9; and chemokines/cytokines such as CCL-18 and epidermal growth factor (EGF) contribute to tumor invasion and metastasis. Angiogenesis: tumor angiogenesis is promoted by the TAM-derived VEGF, IL-8, PDGF, and basic fibroblast growth factor (bFGF), MMPs. Immunosuppression: IL-10 and TGF-β derived from TAMs promote the immunosuppressive activity of regulatory T cells. PD-L1/L2, B7-H4, and VISTA expressed on TAMs trigger the inhibitory PD-1-mediated immune checkpoint in T cells.

**Figure 2 cancers-13-01086-f002:**
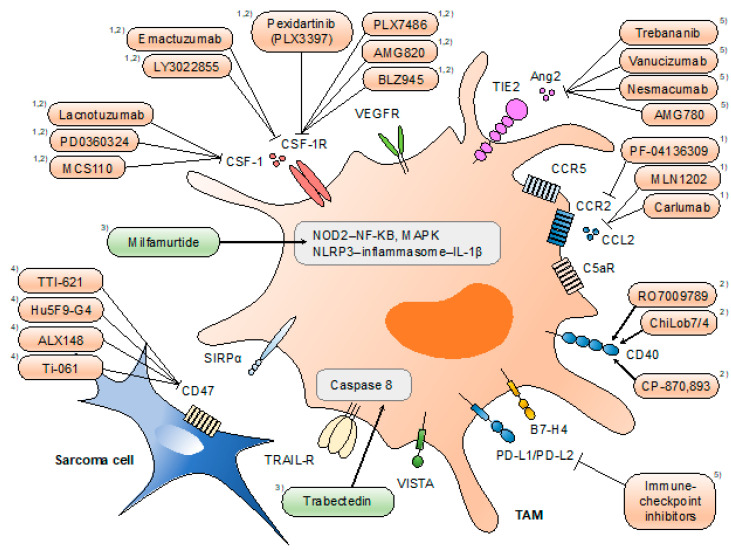
Therapeutic approaches targeting tumor-associated macrophages (TAMs). Strategies directed at TAMs include (1) limiting macrophage recruitment, (2) reprogramming TAMs into antitumor macrophages, (3) targeting the activation of TAMs, (4) activation of macrophage phagocytosis, and (5) others. Drugs highlighted in orange are tested under clinical trials, and those highlighted in green are approved for clinical use (in certain areas). Mifamurtide is indicated in children, adolescents, and young adults for the treatment of high-grade, resectable, non-metastatic osteosarcoma in the EU. Trabectedin partially depletes circulating monocytes and TAMs. Pexidartinib is currently being investigated for single and combined use with sirolimus under clinical phase I/II trials for a variety of histological subtypes of sarcoma.

**Table 1 cancers-13-01086-t001:** Clinicopathological relevance of the infiltration of TAMs in bone sarcomas.

Histological Subtype	Number of Patients	TAM Markers	TAM Density	Clinical Significance of TAMs	Year	Reference
Osteosarcoma	53 (cohort 1) 88 (cohort 2) 20 (cohort 3)	Macrophage: CD14 M1: CD14/HLA-DRα M2: CD14/CD163	Mean number of macrophages: 55 cells per core (non-metastatic disease) 27 cells per core (metastatic disease)	A higher number of CD14^+^ macrophages was associated with better overall survival, vascularity, better response to chemotherapy. M1/M2 phenotype: no correlation with survival CD14^+^ macrophages: correlated with angiogenesis	2011	[36]
Osteosarcoma	50	Macrophage: CD68 M1: INOS M2: CD163 Others: CD3, CD4, CD8, CD20, CD117, CD31, CD146, SMA, OPG	INOS: localized, 3%; metastatic, 0% CD163: localized, 1.2%; metastatic, 0.5% CD146: localized, density score 3 = 0%; metastatic, density score 3 = 7.1%	Polarized macrophages in favor of M1 were associated with non-metastatic process. INOS and OPG: highly correlated with each other CD163 and CD146: highly correlated with each other	2016	[37]
Osteosarcoma	124	Macrophage: CD68, CD163 M2: CMAF M1: pSTST1 Others: CD8, PD-1, PD-L1	CD163: 43.8%, ≥ 50% positive cells CD68: 23.4%, ≥ 50% positive cells	A high level of CD163^+^ macrophages in biopsy specimens significantly correlated with a higher overall survival rate. CD68 and CD163: highly correlated	2017	[38]
Osteosarcoma	247	Macrophage: CD68 M2: CD163 Checkpoints: CD47 (tumor), SIRPα (macrophage)	CD68: 110/mm^2^ (median) CD163: 150/mm^2^ (median) CD47: positive in 53% SIRPα: positive in 32%	CD47 (tumor) and SIRPα (macrophage) expressions showed weak positive correlations with both CD68 and CD163 expressions across all sarcomas. A CD47 (tumor) expression was an adverse prognostic factor in osteosarcoma. A lower SIRPα (macrophage) score was associated with worse overall survival in the non-translocation sarcomas.	2020	[39]
Ewing sarcoma	41	Macrophage: CD68, CD14	CD68 low (≤30 cells/HPF): 51% CD68 high (>30 cells/HPS): 49%	A higher level of CD68^+^ macrophages was associated with poorer overall survival (independent prognostic factor), enhanced vascularity, and increase CRP and WBC counts.	2011	[40]
Ewing sarcoma	24	Macrophage: CD68 M2: CD163	Modest to massive infiltration: CD68: 71% CD163: 79%	A high density of CD163^+^ macrophages was associated with localized disease and longer survival.	2018	[41]

**Table 2 cancers-13-01086-t002:** Clinicopathological relevance of the infiltration of TAMs in soft-tissue sarcomas.

Histological Subtype	Number of Patients	TAM Markers	TAM Density	Clinical Significance of TAMs	Year	Reference
Leiomyosarcoma	149	Macrophage: CD68, CD163	Almost all cases contained either CD68^+^ or CD163^+^ macrophages. Non-gynecologic LMS; CD68: sparse 31%, moderate 34%, dense 35% CD163: sparse 10%, moderate 45%, dense 45%	Non-gynecologic LMS; The densities of CD68^+^ or CD163^+^ TAMs were significantly associated with disease-specific survival: the 5-year disease-specific survivals with the infiltration of dense, moderate, and sparse CD163^+^ TAMs were <40%, 70%, and 100%.	2008	[47]
Leiomyosarcoma	52	CSF1 associated proteins: CD163, CD16, CTSL	CD163: <10 cells/HPF, 6%; ≥10 cells, 11%; ≥20 cells, 17%; ≥45 cells, 51%; unknown, 15% CD16: <10 cells/HPF, 56%; ≥10 cells, 0%; ≥20 cells, 13%; ≥45 cells, 17%; unknown, 13% CTSL: <10 cells/HPF, 32%; ≥10 cells, 25%; ≥20 cells, 4%; ≥45 cells, 17%, unknown, 23%	Gynecologic LMS; The increased immuonstains of CD16^+^, CTSL^+^, and CD163^+^CD16^+^CTSL^+^ were associated with worse outcome.	2011	[48]
Myxoid liposarcoma	78	Macrophage: CD68 M2: CD163	CD68: high (≥100/10 HPF), 81%; low (<100/10 HPF), 19%	Greater CD68^+^ macrophage infiltration (≥100/10 HPF) was associated with poorer overall survival. Higher levels of CD163^+^ M2-TAMs were also associated with poorer overall survival.	2017	[49]
Synovial sarcoma	36	M2: CD163 Others: CD4, CD8, FOXP3, HLA class 1, PDL1, PDL2	CD163: observed in all patients (median, 444 cells/mm^2^)	An increased infiltration of CD163^+^ macrophages was associated with lower infiltration of CD8^+^ and FOXP3^+^ lymphocytes. A higher infiltration of CD163^+^ macrophages indicated a significantly worse overall and progression-free survival (negative independent prognostic factor for overall survival).	2018	[50]
DFSP	10	M2: CD163, CD206 Others: periostin, MMP1, MMP12	Periostin: positive in all tumors at the peripheral area CD163, CD206: positive in all tumors MMP1, MMP12: positive in the CD163^+^ areas	Periostin-MMP1/MMP12 expression on TAMs in the peripheral area could be a possible mechanism of local aggressiveness of DFSP.	2017	[51]
UPS	28	Macrophage: Iba-1 M2: CD163, CD204 Others: CD8	CD163: 670 ± 368/mm^2^ CD204: 479 ± 390/mm^2^	A high density of CD163^+^ and CD204^+^ macrophages tended to be associated with poor overall survival rate.	2018	[52]
UPS	62	Macrophage: Iba-1, CD68 M2: CD163	Iba1^+^ TAMs: 683/mm^2^ CD163^+^ TAMs: 406/mm^2^ Iba1^+^/CD163^+^ TAMs: 78%	A high density of CD163^+^ TAMs was associated with a high AJCC stage. A high percentage of CD163^+^ TAMs were significantly associated with a high FNCLCC grade and decreased overall survival.	2018	[53]
UPS	67	Macrophage: CD68 M2: CD163	CD68: 460/mm^2^ (median) CD163: 512/ mm^2^ (median)	Pleomorphic sarcoma types demonstrated the highest counts of both CD68^+^ and CD163^+^ macrophages, particularly in UPS. A higher proportion of M2-like macrophages than M1-like macrophages was observed, particularly in UPS.	2020	[39]

Abbreviation: CTSL: cathepsin L; DFSP, dermatofibrosarcoma protuberance; UPS, undifferentiated pleomorphic sarcoma.

## Data Availability

Not applicable.

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
