# Peer review of "Role of Tumor-Associated Macrophages in Sarcomas"

_cancers, 2021, doi:10.3390/cancers13051086_

Round 1
Reviewer 1 Report
This is a scientific and succinct review article on the tumor-associated macrophages (TAMs). The latter plays a significant role in the tumor-related inflammation. I would suggest converting this review to a systematic literature search, following guidelines such as PRISMA, or other. A clear methodology should be written and an objective should be set. In this way, the valuable information that this review offers will be presented in a systematic and comparative manner.
Author Response
Answer: I appreciate the comment. We added our methodology as below. Since this review article is not based on a meta-analysis (because of the limited number of the paper showing the sarcoma TAMs) and covers a wide range of basic field (immune system) to the clinical field (cancer therapy), each inclusion/exclusion criteria varies among the category 2–5. We reviewed the English manuscripts’ abstracts for relevance, and all of the relevant articles are included in categories 3–5. Added descriptions are as follows:
(Revised version)
We searched articles published until December 2020 in PubMed using the following terms: “macrophage,” “tumor-associated macrophage,” “sarcoma,” “bone sarcoma,” “soft-tissue sarcoma,” and histological diagnosis terms, such as “osteosarcoma,” “Ewing sarcoma,” “chondrosarcoma,” “leiomyosarcoma,” “liposarcoma,” “undifferentiated sarcoma,” “synovial sarcoma,” and “dermatofibrosarcoma protuberans,” in various combinations. Abstracts of the manuscripts in English were reviewed for relevance. Studies reporting the prognostic value of TAMs in bone and soft-tissue sarcomas were all included. Finally, we searched ClinicalTrials.gov for clinical trials with TAM-targeted drugs.

Reviewer 2 Report
The authors well summarized the functional significance of tumor associated macrophages in sarcoma.
M1/M2 concept is now thought as an oversimplified concept. M1-like or M2-like should be used in the text. Also the words “Pro-inflammatory/anti-inflammatory” should be corrected (Immunity. 2014 Jul 17;41(1):14-20).
Author Response
Answer: I appreciate the comments and agree with them. We used “M1-like” or “M2-like” instead of M1 or M2 and deleted “pro-inflammatory” or “anti-inflammatory” according to the definition described by Murray et al. (Immunity. 2014 Jul 17;41(1):14-20).

Reviewer 3 Report
Fujiwara et al. reviewed the current literature on the role of TAMs in sarcomas. The review is well written, comprehensive and covers many relevant clinical and preclinical trials.
However, minor points should be revised.
1. Please revise, when the prognosis plateaued. In the 1980s (as stated in the abstract l.26) or in the 1990s (as stated in the introduction l.42).
2. "The 5-year survival rates of these cancers (pancreas, 8.7%; lung, 20.7%; gallbladder, 98 27.5%) were lower than those of other cancers" (l.98). This is true in general, however, this is certainly not only due to differences in TAM infiltration but the biology of the respective tumor entities. In the given context, this sounds misleading.
3. "It is characterized histologically by the production of osteoid osteomas by malignant cells and has a variety of histological subtypes, including conventional (osteoblastic, chondroblastic, and fibroblastic types), telangiectatic, small cell, low-grade central, parosteal, periosteal, high-grade surface, and secondary osteosarcoma 3" (l. 102). This sentence is incorrect. Osteoid osteomas is a well-defined, benign tumor entity that is - by all means - not associated with osteosarcomas. Most likely, the authors want to say: "It is characteritzed histologically by the production of osteoid by malignant cells..."
4. "Despite its unknown mechanisms, Ewing’s sarcoma is regarded as an inflammation-related sarcoma because its clinical presentation includes a frequent elevation of white blood cell (WBC) counts and C-reactive protein (CRP) values (l. 160)". To the Ewing's sarcoma community, this might sound like an overstatement. While there was a correlation between the WBC and prognosis, Ewing sarcoma often appears to be "immunologically silent". It is no "specific" feature of Ewing sarcoma to induce systemic inflammation. This is rather a general feature of cancer. Please revise.
5. "Patients with lower levels of CD163-positive macrophage infiltration or high levels of CD8-positive or FOXP3-positive lymphocyte infiltration were associated with better overall survival (l.224)." FoxP3 is the hallmark of regulatory T cells that often contribute to worse survival. I feel, this needs further discussion.
6. Please revise the spelling of "Dermatofibrosarcoma protuberance" in l.227 + 228. It is spelled "protuberans".
7. "UPS, which was formerly known as malignant fibrous histiocytoma, is one of the 238 most common STSs that is classified as a subcategory of undifferentiated sarcoma (l.238)." Both abbreviations (UPS and SPS) have not been clarified in the text so far. Please revise.
Overall, I highly recommend the review to be published in Cancers after minor revision.
Author Response
- Please revise, when the prognosis plateaued. In the 1980s (as stated in the abstract l.26) or in the 1990s (as stated in the introduction l.42).
Answer: I apologize for our inconsistency between the abstract and introduction. According to the meta-analysis by Allison et al., the introduction of systemic chemotherapy in the 1970s–1980s substantially improved the prognosis of patients with osteosarcoma, but the sarcomas’ prognosis has plateaued since the 1990s. I corrected the description in the abstract: “While most of the therapeutic challenges that target sarcoma cells have been unsuccessful, and the prognosis of sarcomas has plateaued since the 1990s, several clinical trials of these strategies yielded promising results and warrant further investigation to determine their translational benefit in sarcoma patients.”
- "The 5-year survival rates of these cancers (pancreas, 8.7%; lung, 20.7%; gallbladder, 98 27.5%) were lower than those of other cancers" (l.98). This is true in general, however, this is certainly not only due to differences in TAM infiltration but the biology of the respective tumor entities. In the given context, this sounds misleading.
Answer: I appreciate and agree with the comment. The results from Jung et al. are interesting, but the paper lacks the discussion from the tumor entities’ biology perspective, as the reviewer suggested. We changed it to the paper from Ni et al., which confirmed the TAM infiltration's negative prognostic significance in a metaanalysis [32].
(Revised version)
In patients with breast cancer, a high density of CD163+ macrophages has been associated with poor histological grade, hormonal receptor negativity, lymph node metastasis, and poor survival outcome1. Zhang et al. developed a meta-analysis with 55 studies, where they evaluated the correlation between the TAM infiltration and clinical prognosis. They showed the adverse effects of TAMs on survival in breast, gastric, bladder, ovarian, oral, and thyroid cancer patients in their results. In contrast, they observed positive effects of TAMs on survival in patients with colorectal cancer 2.
- "It is characterized histologically by the production of osteoid osteomas by malignant cells and has a variety of histological subtypes, including conventional (osteoblastic, chondroblastic, and fibroblastic types), telangiectatic, small cell, low-grade central, parosteal, periosteal, high-grade surface, and secondary osteosarcoma 3" (l. 102). This sentence is incorrect. Osteoid osteomas is a well-defined, benign tumor entity that is - by all means - not associated with osteosarcomas. Most likely, the authors want to say: "It is characteritzed histologically by the production of osteoid by malignant cells..."
Answer: I appreciate the comment. This was our mistake: we wanted to describe that “It is histologically characterized by the production of osteoid by malignant cells and has a variety of histological subtypes, including conventional (osteoblastic, chondroblastic, and fibroblastic types), telangiectatic, small cell, low-grade central, parosteal, periosteal, high-grade surface, and secondary osteosarcoma.” We corrected our wrong description.
- "Despite its unknown mechanisms, Ewing’s sarcoma is regarded as an inflammation-related sarcoma because its clinical presentation includes a frequent elevation of white blood cell (WBC) counts and C-reactive protein (CRP) values (l. 160)". To the Ewing's sarcoma community, this might sound like an overstatement. While there was a correlation between the WBC and prognosis, Ewing sarcoma often appears to be "immunologically silent". It is no "specific" feature of Ewing sarcoma to induce systemic inflammation. This is rather a general feature of cancer. Please revise.
Answer: I appreciate the comment. Accordingly, I deleted the original description and added the sentence as follows:
(Revised version)
Despite its unknown mechanisms, increased white blood cell (WBC) counts, elevated C-reactive protein (CRP) concentration, and increased erythrocyte sedimentation rate (ESR) are frequently observed.
- "Patients with lower levels of CD163-positive macrophage infiltration or high levels of CD8-positive or FOXP3-positive lymphocyte infiltration were associated with better overall survival (l.224)." FoxP3 is the hallmark of regulatory T cells that often contribute to worse survival. I feel, this needs further discussion.
Answer: I appreciate and agree with the comment. Regarding the association between a higher FOXP3’s infiltration and patients’ better overall survival, Oike et al. described that their results differed from those in former publications, showing the opposite results 3. One possible explanation was that FOXP3+ lymphocytes infiltrated into the tumor microenvironment in SS and might represent a subset other than T-reg 4. This is because FOXP3+ T cells comprise functionally different subsets, including non‐T-reg. A second explanation was that a favorable prognosis might have resulted from their choice of a cut-off value (a median value) different from Que et al.’s former publication, where the cut-off value was 5% of tumor-stromal cells3, 4. Further studies are needed to determine the role of FOXP3+ T cells in SS. We added this discussion as follows:
(Revised version)
Regulatory T cells promote evasion of cancer cells from immune responses and often contribute to worse survival 5, 6. Thus, the correlation between a high infiltration of FOXP3+ lymphocytes and a better survival outcome in SS was different from the former publications 4. This result may be due to their choice of cut-off value and their assumption that FOXP3+ lymphocytes infiltrate into the tumor microenvironment in SS might represent a subset other than regulatory T cells 4. This is because FOXP3+ T cells comprise functionally different subsets, including non‐regulatory T cells. Further studies are needed to determine the role of FOXP3+ T cells in SS.
- Please revise the spelling of "Dermatofibrosarcoma protuberance" in l.227 + 228. It is spelled "protuberans".
Answer: I appreciate the comment. I modified it to “dermatofibrosarcoma protuberans.”
- "UPS, which was formerly known as malignant fibrous histiocytoma, is one of the 238 most common STSs that is classified as a subcategory of undifferentiated sarcoma (l.238)." Both abbreviations (UPS and SPS) have not been clarified in the text so far. Please revise.
Answer: I appreciate the comment. I clarified “UPS” and corrected it to “STSs” to “soft-tissue sarcomas.” The revised sentence is as follows:
(Revised version)
Undifferentiated pleomorphic sarcoma (UPS), formerly known as malignant fibrous histiocytoma, is one of the most common soft-tissue sarcomas classified as a subcategory of undifferentiated sarcoma 3.

Reviewer 4 Report
This is a solid and informative review. Fujiwara et al. comprehensively summarize results regarding the role of TAMs in different sarcoma subtypes including the overview of clinically promising agents to target TAMs in sarcomas. However, there are several formal issues and inconsistencies among figures and text (as listed below) which should be addressed to improve the overall quality of the review.
MAJOR COMMENTS:
* Figure 1: A drawing of monocyte is missing? In addition, the morphology of TAM is odd, bean-like shape? Based on the strange placement of the "TAM" label, I suppose that this figure suffers some serious formatting issues. I would also suggest re-creating Figure 1 in the same style as is used in Figure 2, i.e., to use the same style of shading and black outlines of objects to make the figures more consistent. Furthermore, red lines from TAM should be moved closer to the TAM. Sarcoma cell morphology should be a little bit more realistic. This is not a typical sarcoma (or mesenchymal-like) cell morphology.
* Figure 2: About a half of agents depicted in this figure is not discussed in the manuscript nor referenced. Some targets, i.e. TIE2/Ang2, are also completely omitted in the text. The image resolution of this figure is poor, which suggests that the image might be reused and not prepared specifically for this review (?). Either way, the information presented in Figure 2 should be supported in the text/figure legend and properly referenced (or omitted for clarity). In addition, Lines 265-269 describe classification of TMA-directed therapeutic strategies and refer to Figure 2. However, Figure 2 does not show or follow this classification which is also confusing.
* Please put citations right after you first cite from another source. Citations are sometimes too far from the first sentence (e.g., Line 143 vs 155; Line 165 vs 168). Please also check and amend where references should be repeated for clarity (e.g., Lines 198-99: not clear whether “these data” refers to Ref [44] or 43] or both).
MINOR COMMENTS:
* A full protein name followed by its symbol in parentheses should be provided for all important proteins (such as CSF-1, CCL2, IL-6 etc.,) upon first usage, in a similar manner as presented at Line 76: „matrix metalloproteinase (MMP)-2“. Some proteins are written out in the last section 5 (Lines 302-305), while their abbreviations are already used in section 2.
* Section “4.3” is used twice (Synovial sarcoma, Line 215; Dermatofibrosarcoma protuberance, Line 227)
* I would recommend avoiding all unnecessary abbreviations for sarcoma subtypes that are mentioned only few times in the manuscript, e.g., MLS (section 4.2), SS (section 4.3), DFSP (section 4.3, Line 228), TGCT (Line 290).
* Inconsistent usage of “CD163+” vs. “CD163-positive” throughout the manuscript. If there is no difference in the meaning, I would recommend unifying the naming and apply it wherever referring to cells positive for expression of a CD molecule.
* Line 43: Please include some examples of the most common soft-tissue sarcomas in this sentence.
* Line 94-95: Meaning not clear, please rephrase.
* Line 117 vs 120: Inconsistency in “HL-DRA” vs “HLA-DRalpha”
* Line 123-137: Contradictory results should be presented clearly as such.
* Line 141-142 - CD47/SIRPalpha axis needs more detailed explanation and a reference to help readers to understand the results of Duncsok et al. discussed later in that paragraph.
* Line 164-180: Again, contradictory results presented in a way that gives the impression that there is a consensus between the studies and that their results provide incremental understanding of the role of M2 in Ewing's sarcoma. However, the exact opposite is true, and this should be clearly stated in the review.
* Table 2 - Ref [44] the percantages for individual categories (CD163do not make 100% when combined.
* Line 278: “as part of a clinical trial.“ - Please provide the clinical trial registration number or an appropriate reference.
Author Response
* Figure 1: A drawing of monocyte is missing? In addition, the morphology of TAM is odd, bean-like shape? Based on the strange placement of the "TAM" label, I suppose that this figure suffers some serious formatting issues. I would also suggest re-creating Figure 1 in the same style as is used in Figure 2, i.e., to use the same style of shading and black outlines of objects to make the figures more consistent. Furthermore, red lines from TAM should be moved closer to the TAM. Sarcoma cell morphology should be a little bit more realistic. This is not a typical sarcoma (or mesenchymal-like) cell morphology.
Answer: I appreciate the comment. According to the comment, I modified Figure 1, including a monocyte drawing, the same style as Figure 2 with shading and black outlines of objects and red lines closer to the TAM. We drew the TAMs with a spindle shape, typically observed in in vitro and histological evaluations. Besides, I corrected the shape of a sarcoma cell to the spindle one, typical morphology of most of the sarcoma subtypes and different from TAMs (of course). We placed the TAM between M1-like macrophage and M2-like macrophage but closer to M2-like macrophage to show that the TAMs exhibit an M2‐like profile 7.
* Figure 2: About a half of agents depicted in this figure is not discussed in the manuscript nor referenced. Some targets, i.e., TIE2/Ang2, are also completely omitted in the text. The image resolution of this figure is poor, which suggests that the image might be reused and not prepared specifically for this review (?). Either way, the information presented in Figure 2 should be supported in the text/figure legend and properly referenced (or omitted for clarity). In addition, Lines 265–269 describe classification of TMA-directed therapeutic strategies and refer to Figure 2. However, Figure 2 does not show or follow this classification which is also confusing.
Answer: I appreciate the comment. In this review, we focused on the promising or currently available drugs for sarcomas, and their details are described in the text/figure legend, omitting the details of drugs that are tested for other malignancies for clarity. However, I agree with the additional information in Figure 2 should be supported. We added the information and related references in the revised version (as below). Also, we amended Figure 2 to follow the classifications in the text. Of note, we created Figure 2 specifically for this review. We believe the problem of the image resolution will be resolved under further editorial process.
(Revised version)
The other candidate targets of TAMs include CCL2-CCR2, CD40, and CD47-SIRPα 11, 12, 75.
CD40 is a surface marker of macrophages, which is highly expressed on M1-like TAMs 79. The combination of a CD40 agonist with gemcitabine effectively circumvented tumor-mediated immune suppression by promoting antitumor macrophage, which increased the survival rate in patients with surgically incurable pancreatic ductal adenocarcinoma (PDAC) 80. Several phase I trials of the anti-CD40 agonist have shown tolerability (NCT01433172) 81, 82
The angiopoietin receptor TIE2, a molecule previously known to be restricted to endothelial and hematopoietic stem cells 8, 9, is expressed on a subset of TAMs 10. TIE2+ TAMs are closely associated with tumor vasculature and have been crucial for angiogenesis, which depends on angiopoietin-2 (Ang2), a TIE2’s ligand produced by endothelial cells. Several drugs designed to target the Ang2–TIE2 axis, such as trebananib and venucizumab, have been tested for solid tumors 11, 12. Trebananib was combined with paclitaxel, trastuzumab, or bevacizumab, which was tolerable but, so far, its efficacy has been limited in patients with breast cancer 13. Venucizumab has an acceptable safety and tolerability profile for a heterogeneous group of advanced solid tumors 14, but combined use with mFOLFOX-6 showed limited clinical benefit for metastatic colorectal cancer 15.
PD-1/PD-L1 axis has been an attractive target in cancer immunotherapy. PD-1 expression on TAMs correlates negatively with phagocytic potency against tumor cells 16, raising a possible effect of the combination of macrophage-targeted therapy and immune checkpoint modulation. A preclinical study demonstrated that a combination of CSF-1R inhibitor with PD-1 or CTLA4 antagonists elicited tumor regression, while single use of PD-1 or CTLA4 inhibitors showed limited efficacy 17. However, a phase I/IIa trial of PLX3397 with pembrolizumab, an anti-PD-1 antibody, for advanced melanoma and other solid tumors recently terminated for insufficient clinical efficacy (NCT02452424).
* Please put citations right after you first cite from another source. Citations are sometimes too far from the first sentence (e.g., Line 143 vs 155; Line 165 vs 168). Please also check and amend where references should be repeated for clarity (e.g., Lines 198-99: not clear whether “these data” refers to Ref [44] or 43] or both).
Answer: I appreciate the comment. “These data” in lines 198–199 refers to Ref [44]. I checked and amended the citations to the appropriate place throughout the manuscript.
MINOR COMMENTS:
* A full protein name followed by its symbol in parentheses should be provided for all important proteins (such as CSF-1, CCL2, IL-6 etc.,) upon first usage, in a similar manner as presented at Line 76: „matrix metalloproteinase (MMP)-2“. Some proteins are written out in the last section 5 (Lines 302-305), while their abbreviations are already used in section 2.
Answer: I appreciate the comment. I added the full protein names followed by its symbol in the parenthesis throughout the manuscript. I also deleted the unnecessary protein names whose symbols I had already used.
* Section “4.3” is used twice (Synovial sarcoma, Line 215; Dermatofibrosarcoma protuberance, Line 227)
Answer: Thank you for the comment. I revised the number accordingly.
* I would recommend avoiding all unnecessary abbreviations for sarcoma subtypes that are mentioned only few times in the manuscript, e.g., MLS (section 4.2), SS (section 4.3), DFSP (section 4.3, Line 228), TGCT (Line 290).
Answer: Thank you for the comment. Accordingly, I avoided all unnecessary abbreviations that are mentioned only a few times throughout the manuscript.
* Inconsistent usage of “CD163+” vs. “CD163-positive” throughout the manuscript. If there is no difference in the meaning, I would recommend unifying the naming and apply it wherever referring to cells positive for expression of a CD molecule.
Answer: I appreciate the comment. I unified the term to “CD163+” throughout the manuscript.
* Line 43: Please include some examples of the most common soft-tissue sarcomas in this sentence.
Answer: Thank you for the comment. I included the examples in this sentence.
* Line 94-95: Meaning not clear, please rephrase.
Answer: Thank you for the comment. I rephrased this sentence as follows:
(Revised version) The common markers for M1 TAMs in human samples are human leucocyte antigen (HLA)-DR, inducible nitric oxide synthase (iNOS), and pSTAT1. On the other hand, common markers for M2 TAMs are CD163, CD204, and CD206, attributable to the high expression of the mannose receptor-1 (CD206) and macrophage scavenger receptors (CD163 and CD204) by the M2 TAMs.
* Line 117 vs 120: Inconsistency in “HL-DRA” vs “HLA-DRalpha”
Answer: I appreciate the comment. I corrected it to “HLA-DRα.”
* Line 123-137: Contradictory results should be presented clearly as such.
Answer: I appreciate the comment. I observed contradictory results in Dumars et al.’s and Gomez-Brouchet et al.’s reports. I amended the original description to show this difference as follows:
(Revised version) Controversy exists regarding the prognostic significance of the M1/M2-phenotype in osteosarcoma. Dumars et al. compared the expressions of several molecules, including TAM markers, … Dumars et al. concluded that a dysregulation of M1/M2 polarization in favor of M1-TAMs is associated with localized osteosarcoma 36. On the contrary, the correlation between the M2-phenotype and worse prognosis was demonstrated by Gomez-Brouchet et. al 37.
* Line 141-142 - CD47/SIRPalpha axis needs more detailed explanation and a reference to help readers to understand the results of Duncsok et al. discussed later in that paragraph.
Answer: I appreciate the comment. I added more detailed explanation and references in the revised version as follows:
(Revised version) The clinical significance of CD47 and signal-regulatory protein α (SIRPα), which are macrophage-related checkpoints, were recently reported. CD47, a transmembrane protein found ubiquitously expressed on normal cells, has increased its expression in a high proportion of malignant tumor cells 18, 19. This protein acts primarily as a dominant “don't eat me” signal. If the tumor cells express CD47, it binds to SIRPα on phagocytic immune cells, preventing engulfment 18-20.
* Line 164-180: Again, contradictory results presented in a way that gives the impression that there is a consensus between the studies and that their results provide incremental understanding of the role of M2 in Ewing's sarcoma. However, the exact opposite is true, and this should be clearly stated in the review.
Answer: I appreciate the comment. I revised the descriptions to show the opposite results clearly as follows:
(Revised version) On the other hand, Handl et al. showed no correlation between CD68+ macrophage density and clinicopathological parameters 41. However, the higher number and density of CD163+ TAMs were correlated with localized disease, and there was a trend toward more prolonged survival in relation to a higher density of CD163+ TAMs 41. In this study, 71% and 79% of 24 cases showed a modest to massive infiltration of CD68+ and CD163+ cells, respectively 41. Further analyses based on the larger cohorts are necessary to clarify the prognostic roles of CD68+ and CD163+ macrophage density in Ewing’s sarcoma.
* Table 2 - Ref [44] the percantages for individual categories (CD163do not make 100% when combined.
Answer: I appreciate the comment. According to the Ganjoo et al (44), the percentages for TAM markers were described as follows: CD163: <10 cells/HPF, 6%; ≥10 cells, 11%; ≥20 cells, 17%; ≥45 cells, 51%; unknown, 15%; CD16: <10 cells/HPF, 56%; ≥10 cells, 0%; ≥20 cells, 13%; ≥45 cells, 17%; unknown, 13%; CTSL: <10 cells/HPF, 32%; ≥10 cells, 25%; ≥20 cells, 4%; ≥45 cells, 17%, unknown, 23%. The original version excluded the percentages categorized as “unknown.” I included them in the revised version, which make 100% when combined.
* Line 278: “as part of a clinical trial.“ - Please provide the clinical trial registration number or an appropriate reference.
Answer: Thank you. I added the clinical trial registration number: NCT01004861, NCT01525602, NCT02390752, and NCT02584647.
